# Self-organization of channels and hillslopes in models of fluvial landform evolution and its potential for solving scaling issues

Stefan Hergarten[1] and Alexa Pietrek[1]

[1]Institut für Geo- und Umweltnaturwissenschaften, Albert-Ludwigs-Universität Freiburg, Albertstr. 23B, 79104 Freiburg, Germany

**Correspondence:** Stefan Hergarten
(stefan.hergarten@geologie.uni-freiburg.de)

**Abstract.** Including hillslope processes in models of fluvial landform evolution is still challenging. Since applying the respective models for fluvial and hillslope processes to the entire domain causes scaling problems and makes the results dependent on the spatial resolution, the domain is explicitly subdivided into channels and hillslopes in some models. The transition from hillslopes to channels is typically attributed to a given threshold catchment size as a proxy for a minimum required discharge. Here we propose a complementary approach for delineating channels based on the discrete representation of the topography. We assume that sites with only one lower neighbor are channelized. In combination with a suitable model for hillslope processes, this concept initiates a self-organization of channels and hillslopes. A numerical analysis with a simple model for hillslope dynamics reveals no scaling issues, so that the results appear to be independent of the spatial resolution. The approach predicts a break in slope in the sense that all channels are distinctly less steep than hillslopes. On a regular lattice, the simple D8 flow routing scheme (steepest descent among the 8 nearest and diagonal neighbors) harmonizes well with the concept proposed here. The D8 scheme works well even when applied to the hillslopes. This property simplifies the numerical implementation and increases its efficiency.

## 1  Introduction

Models of the stream-power type have been successfully applied in modeling fluvial landform evolution at large scales for a long time (for an overview, see, e.g., Coulthard, 2001; Willgoose, 2005; Wobus et al., 2006; van der Beek, 2013). Instead of simulating the processes in a river in detail, these models describe the long-term contribution of river segments to landform evolution based on strongly simplified relations. The stream-power incision model (SPIM) is the simplest model of this type. It predicts the erosion rate $E$ as a function of the upstream catchment size $A$ (a proxy for the mean discharge) and the channel slope $S$ in the form

$$E = K A^m S^n. \tag{1}$$

The SPIM involves only three parameters, $K$, $m$, and $n$. The ratio of the exponents $m$ and $n$ is constrained quite well by long profiles of real-world rivers. Hack (1957) found the relation

$$S \propto A^{-\theta}, \tag{2}$$

where $\theta$ is called the concavity index. This relation has been investigated in numerous studies, whereby nowadays either $\theta = 0.45$ or $\theta = 0.5$ is typically used as a reference value (e.g., Whipple et al., 2013; Lague, 2014). Interpreting Eq. (2) as the fingerprint of spatially uniform erosion yields $\frac{m}{n} = \theta$. The absolute values of $m$ and $n$ are, however, more uncertain (e.g., Lague, 2014; Harel et al., 2016; Hilley et al., 2019; Adams et al., 2020). The widely used choice $n = 1$ is mainly a matter of convenience since the model is linear with regard to the channel slope $S$ (and thus also with regard to the surface elevation) then. The third parameter, $K$, is called the erodibility. It is a lumped parameter that summarizes all influences on erosion beyond catchment size and channel slope.

The SPIM implements the concept of detachment-limited erosion in the sense that all particles entrained by the river are immediately swept out of the system. This means that the effect of sediment transport on landform evolution is completely disregarded. Owing to this limitation, the SPIM is rather a tool for understanding and analyzing some fundamental properties of rivers than a general model of fluvial landform evolution. In turn, the numerical landform evolution models reviewed by Coulthard (2001), Willgoose (2005), and van der Beek (2013) as well as more recent developments such as Cidre (Carretier et al., 2016) and SPACE (Shobe et al., 2017) contain a sediment balance.

In this field, the linear decline model (Whipple and Tucker, 2002), the $\xi$–$q$ model (Davy and Lague, 2009), and the shared stream-power model (Hergarten, 2020b) are remarkably simple. Mathematically, the three concepts are even equivalent and involve only one additional parameter compared to the SPIM. In this study, the shared stream-power model is used as an example of a simple model of fluvial erosion and sediment transport. It is described by the equation

$$\frac{E}{K_d} + \frac{Q}{K_t A} = A^m S^n, \tag{3}$$

where $Q$ is the sediment flux (volume per time). While the SPIM (Eq. 1) uses a single lumped parameter for the erodibility, the shared stream-power model involves two parameters $K_d$ and $K_t$ with the same physical units. The parameter $K_d$ describes the ability to erode the river bed, while the transport capacity

$$Q_c = K_t A^{m+1} S^n \tag{4}$$

(the sediment flux at $E = 0$) is proportional to $K_t$.

While the the equation for the change in surface elevation $H$ at a given uplift rate $U$ is the same as for the SPIM (and other models in this context),

$$\frac{\partial H}{\partial t} = U - E, \tag{5}$$

taking into account sediment transport requires an additional balance equation. Assuming that each node $i$ of a discrete grid delivers its entire sediment flux $Q_i$ to a single neighbor, the sediment balance equation reads

$$E_i = \frac{Q_i - \sum_j Q_j}{s_i}, \tag{6}$$

where $s_i$ is the size (area) of the respective grid cell. The right-hand side of Eq. (6) is a discrete representation of the divergence operator with the sum extending over all neighbors $j$ that deliver their sediment flux to the cell $i$.

Since the shared stream-power model only serves as an example in this study, only its most important properties are described in the following, and readers are referred to previous work (Hergarten, 2020b, 2021). The model turns into the SPIM for $K_t \to \infty$ and into a transport-limited model for $K_d \to \infty$. For spatially uniform erosion, the sediment flux is $Q = EA$, and Eq. (3) collapses to a form analogous to the SPIM (Eq. 1) with an effective erodibility $K$ according to

$$\frac{1}{K} = \frac{1}{K_d} + \frac{1}{K_t}. \tag{7}$$

Therefore, equilibrium topographies under uniform uplift depend only on $K$, but not on the individual values $K_d$ and $K_t$. In particular, the channel slope is

$$S = \left(\frac{E}{KA^m}\right)^{\frac{1}{n}}. \tag{8}$$

Considerable progress was recently made concerning the numerical treatment of the shared stream-power model and the respective mathematically equivalent models (Yuan et al., 2019; Hergarten, 2020b). In particular, the fully implicit scheme for 65 the linear model ($n = 1$) proposed by Hergarten (2020b) achieves almost the same performance as the implicit scheme for the SPIM (Hergarten and Neugebauer, 2001; Braun and Willett, 2013). The main aspect where these models are still more complicated than the SPIM is the need to consider the entire topography including the hillslopes. While the SPIM can be applied to individual channels or channel networks, all models that involve a sediment balance require the sediment flux from the hillslopes into the rivers.

As long as the spatial resolution is low (typically some hundred meters), fluvial processes may be dominant over hillslope processes even down to the pixel scale. Then the fluvial model may be applied to all sites without taking into account hillslope processes explicitly. At higher resolutions, however, Eq. (2) predicts an increase in equilibrium channel slope towards drainage divides since the minimum catchment size is defined by one grid cell. This finally leads to steep walls at drainage divides. In order to avoid the occurrence of such unrealistic topographies, models of fluvial landform evolution need to be extended by 75 hillslope processes, where the linear diffusion equation (Culling, 1960) is the simplest model. The diffusion model assumes a sediment flux per unit length of

$$\boldsymbol{q} = -D\nabla H, \tag{9}$$

where $D$ is the diffusivity and $\nabla$ the 2-D gradient operator. Diffusion is added to a landform evolution model by adding the negative divergence of $\boldsymbol{q}$ to the right-hand side of Eq. (5).

However, simply applying models of fluvial erosion and hillslope processes to all sites causes scaling problems. To our knowledge, these problems have been investigated systematically only for the specific combination of the SPIM with diffusion (Perron et al., 2008; Pelletier, 2010; Hergarten, 2020a). There, the primary problem arises from combining the sediment flux (volume per unit time) from the hillslopes into the rivers with the fluvial erosion rate (Eq. 1). Combining these properties requires a finite area over which erosion acts. Simply considering channels on a pixel-by-pixel basis would make the results 85 strongly dependent on the cell size of the grid. This issue can be solved by assigning a finite width to each channel and

assuming that erosion only concerns a part of each cell, as already proposed by Howard (1994) for a more comprehensive model. Hergarten (2020a) proposed a slightly different concept, but the effect is finally similar.

However, (Hergarten, 2020a, Fig. 10) observed a residual dependence on grid spacing even after rescaling the parameters accordingly. This effect is owing to the transition between hillslope processes and fluvial erosion, in particular to the occurrence of parallel flow patterns in regions where fluvial erosion still has a considerable effect. Since catchment sizes depend on grid spacing for parallel flow patterns, fluvial erosion depends on the spatial resolution in the transition zone. As transport capacities (e.g., Eq. 4) also depend on the spatial resolution, this issue is not exclusive to the SPIM.

Since contemporary large-scale modeling studies typically use spatial resolutions of some 100 meters, the resulting scaling issue is hardly visible. For typical diffusivities in an order of magnitude of $0.01 \mathrm{~m^2 yr^{-1}}$ (e.g., Godard et al., 2013), the effect of diffusion on the flow pattern is negligible at the grid scale. However, the scaling problem may be revealed if parameter values are varied over some orders of magnitude. As an example, Godard et al. (2013) considered the response of sediment fluxes to climatic oscillations with the model CHILD (Tucker et al., 2001). Investigating the relation between erodibility, diffusivity, frequency, and amplitude, they found deviations in the exponents from the theoretically predicted values. Such deviations in exponents point towards influences beyond the model parameters, which may be the grid spacing. An example of such an effect will be shown at the end of Sect. 2.

The problem arising from applying a fluvial erosion model for channels to parallel flow patterns can be circumvented by separating channels from hillslopes. Willgoose et al. (1991) introduced a continuous channel indicator function for a smooth transition from hillslope processes to fluvial erosion. As a simpler concept, defining a threshold catchment size $A_\mathrm{c}$ and separating the domain accordingly into hillslope ($A < A_\mathrm{c}$) and channel sites ($A \geq A_\mathrm{c}$) has also been used (e.g., Campforts et al., 2017).

However, there is no universal value for such a threshold $A_\mathrm{c}$ since it depends on the involved processes and on their parameters. As an example, hillslope diffusion smoothens the topography and thus counteracts the formation of channels. Therefore, $A_\mathrm{c}$ should increase with increasing diffusivity. Instead of introducing an additional model for $A_\mathrm{c}$ based on the involved processes, leaving the decision to the landform evolution model would be more elegant and probably also more robust. This would be a self-organization of channels and hillslopes without any explicit forcing by a threshold. We will see in Sect. 2 that applying fluvial erosion and diffusion to all sites already allows for such a self-organization, but exhibits unreasonable scaling properties.

Developing a concept for the self-organization of channels and hillslopes based on the processes acting in the two domains is the subject of this study. The task comprises two steps. In the following section, we introduce a simple scheme for delineating channels on a given topography without defining a threshold catchment size explicitly. Afterwards, we attempt to specify the requirements to the processes acting in channels and on hillslopes that enable such a self-organization in combination with a consistent scaling behavior.

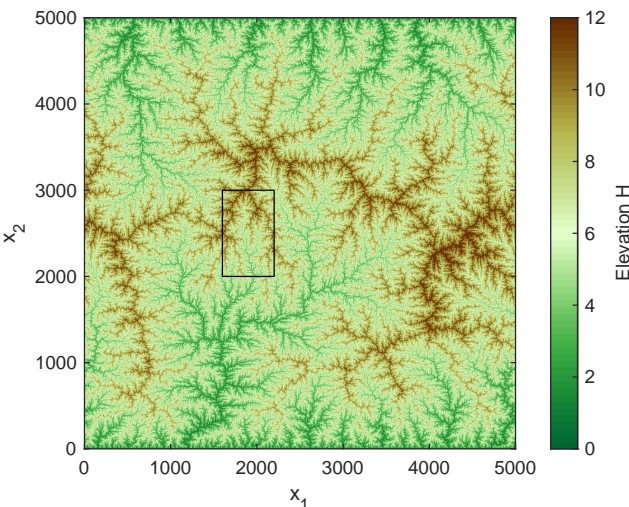

**Figure 1.** Fluvial equilibrium topography on a $5000 \times 5000$ grid obtained by Hergarten (2020b). The rectangle defines the region considered in Figs. 3 and 4.

## 2 A simple criterion for delineating channels

The simplest scheme of flow routing on a given topography assumes that the discharge of each cell is entirely delivered to one of its neighbors. This neighbor is typically selected by the steepest-descent criterion, so by the maximum ratio of elevation drop and horizontal distance. This ratio also defines the channel slope $S$. On regular meshes, the D8 flow routing scheme (O'Callaghan and Mark, 1984) taking into account the eight nearest and diagonal neighbors is widely used. In turn, more elaborate flow routing schemes such as the MFD (multiple flow directions) scheme (Freeman, 1991; Quinn et al., 1991) or the D∞ scheme (Tarboton, 1997) are able to distribute the discharge among multiple neighbors.

Instead of introducing a minimum catchment size as a criterion for channelized flow, we simply define sites that have only one neighbor with a lower elevation as channel sites. For such sites, the D8 scheme (or an equivalent single-flow direction scheme on an irregular grid) would capture the flow direction well, and schemes using multiple neighbors would not yield a different result. This concept reflects the idea that a thin layer of water is focused into one direction without spreading laterally.

As a second rule for delineating channels, we assume that the flow target of a channel site is also a channel site even if it has more than one lower neighbor. This means that a channel never turns into distributed flow. While this rule is not relevant for the examples considered in this study, it may become important for rivers in a rather flat, tectonically inactive foreland region (e.g., Hergarten, 2022a).

As a first test, we apply this concept to synthetic topographies. The first topography is a fluvial equilibrium topography under uniform uplift computed on a $5000 \times 5000$ grid for $m = 0.5$ and $n = 1$ in nondimensional coordinates ($K = 1$, $U = 1$) with unit grid spacing. This topography was also used by Hergarten (2020b) and Hergarten (2021) and is shown in Fig. 1. The respective nondimensional coordinates will be used for all simulations throughout this study.

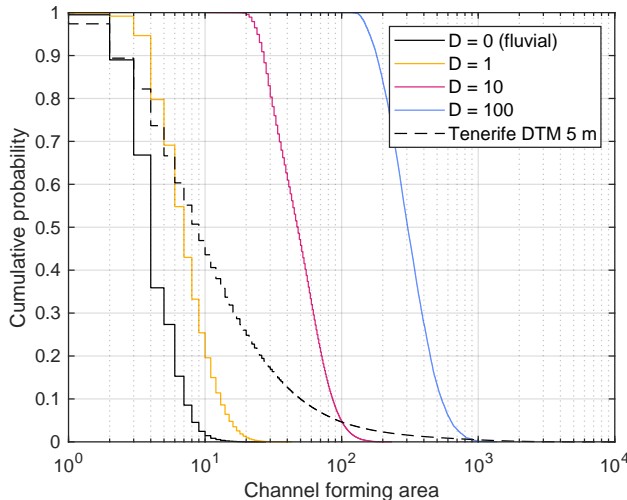

**Figure 2.** Empirical cumulative distributions of the channel forming areas obtained from four synthetic topographies and a DTM of Tenerife. Channel forming area refers to the catchment size of the detected channel heads, measured in pixels.

Figure 2 shows the cumulative distribution of the channel forming areas (the catchment sizes of all channel heads) obtained by our criterion. It is immediately recognized that almost none of the detected channel heads are single-pixel catchments ($A = 1$), although the topography was completely shaped by fluvial erosion. This result is owing to the high channel slope of single-pixel catchments ($S = 1$ here), which makes it unlikely that only one out of the eight neighbors is lower than the respective node. In this case, 7 out of the 8 neighbors must be higher than the considered site, but none of them may drain towards this site. The most frequent channel forming area is 4 pixels (more than 30 % of all channel heads). More than 95 % of all channel heads have a catchment size $A \leq 9$. So the simple concept for delineating channels is not able to recognize the fluvial characteristics of the entire topography, but detects larger channels ($A \gtrsim 10$) quite well.

For comparison, the colored curves in Fig. 2 show the results obtained from the respective topographies with transport-limited fluvial erosion and linear diffusion applied to the entire domain. The respective topographies are shown in Fig. 3. For clarity, only the part of the domain referring to the black rectangle in Fig. 1 is shown. While a diffusivity of $D = 1$ causes only a moderate shift towards larger channel forming areas, increasing the diffusivity to $D = 10$ and to $D = 100$ has a strong effect. For $D = 100$, channel initiation takes place at catchment sizes of several hundred pixels. This result aligns well with the visual impression of smoothing the topography progressively with increasing diffusivity (Fig. 3).

As a real-world example, the 5 m DTM of the Tenerife island (CNIG, 2022) is considered (dashed line in Fig. 2). The limited applicability of our definition to real-world topographies becomes visible here. There are indeed channel heads with a catchment size of several hundred pixels, but more than 50 % of all channel heads have catchments sizes $A \leq 10$, corresponding to $250\,\mathrm{m}^2$. The most frequent channel forming area is even the same as for the artificial fluvial topography (4 pixels or $100\,\mathrm{m}^2$) and thus much to small for real channels.

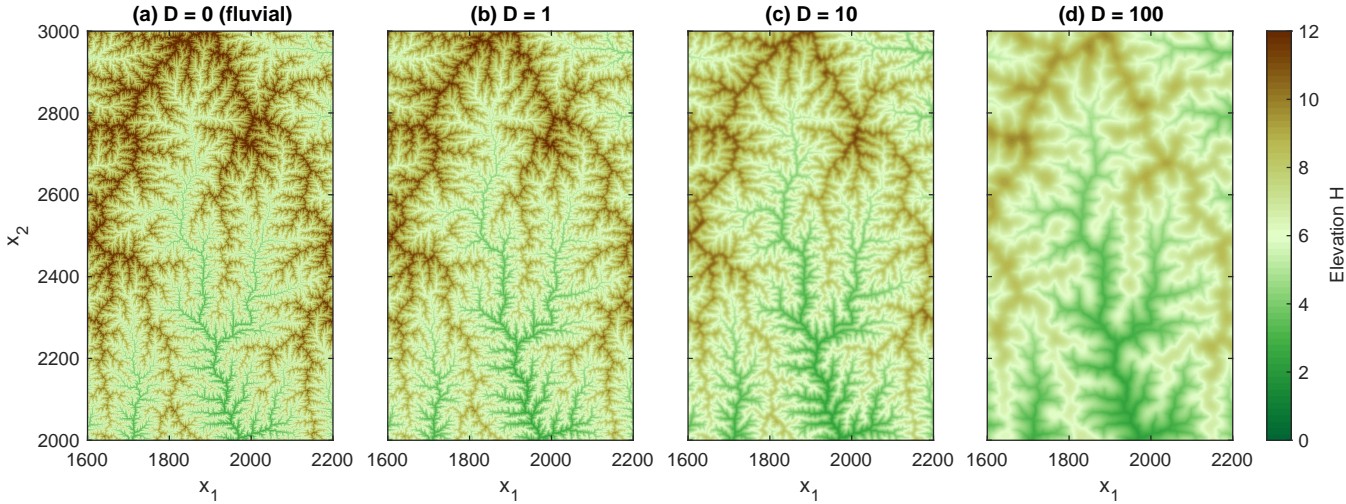

**Figure 3.** Part of the topography defined by the rectangle in Fig. 1 for different values of the diffusivity $D$.

These results suggest that the simple scheme for delineating channels without an additional threshold is unsuitable for application to real-world terrain models, while it may be useful in the context of modeled topographies. With regard to earlier work (e.g., Tribe, 1992), the lack of applicability to real-world topographies is not surprising. In that study, several problems were discussed, and solving them required a much more elaborate approach involving adjustable parameters. Combining such an approach with a simple landform evolution model would be questionable concerning the complexity and the number of parameters. In this sense, developing a simple scheme particularly for landform evolution modeling is useful, regardless of its applicability to real-world topographies.

The results obtained for different diffusivities can also be used for illustrating the scaling problem inherent to the combination of fluvial erosion and diffusion at all sites. The simple model considered here involves only two parameters (except for the uplift rate $U$, which just affects the vertical scale). For $m = 0.5$ and $n = 1$, the unit of the erodibility $K$ is $\mathrm{yr}^{-1}$, so that the fluvial model without diffusion contains no characteristic horizontal length scale. This means that a purely fluvial topography could be rescaled horizontally by any factor, as pointed out by Kwang and Parker (2017). Since the unit of $D$ is $\mathrm{m}^2\mathrm{yr}^{-1}$, diffusion introduces a horizontal length scale. This horizontal length scale is readily obtained from the units of $K$ and $D$ as $\sqrt{\frac{D}{K}}$. Thus, horizontal lengths obtained from simulations with different diffusivities at constant $K$ should be proportional to $\sqrt{D}$, and areas should be proportional to $D$.

However, the cumulative distributions of the channel forming areas shown in Fig. 3 reveal that this in not the case. The distributions are similar concerning their shape, but an increase in $D$ by a factor of 10 results in an increase in channel forming area only by a factor of about 6.5. So the channel forming area increases rather like $D^{0.8}$ than like $D$. This is an example of a scaling relation that deviates from the theoretical prediction, as it was found in a different context by Godard et al. (2013). In principle, a transfer from nondimensional coordinates to real-world properties based on the model parameters is impossible then. In our example, the relation between channel forming area and diffusivity would involve $\mathrm{m}^{1.6}$ at one side and $\mathrm{m}^2$ at

the other side, and there is no way to make the relation dimensionally consistent without taking into account the grid spacing explicitly.

## 3 Self-organization of drainage networks

Starting from the seminal work of Horton (1945) and Hack (1957), scale-invariant properties of river networks have been investigated extensively. The concept of optimal channel networks (OCNs) introduced in the 1990s (Howard, 1990; Rodriguez-Iturbe et al., 1992a, b; Rinaldo et al., 1992, 1998) turned out to be particularly successful in this context. It relies on the idea that drainage networks in an equilibrium between uplift and erosion self-organize towards a state that minimizes the energy dissipated by the water.

However, explaining scale-invariant properties of river networks is not immediately helpful in the context of hillslopes. In turn, looking at the conditions under which this concept predicts networks with realistic properties may provide an idea how to construct a model with self-organizing channels and hillslopes. So let us briefly recapitulate the theory of minimum energy dissipation in river networks. If we neglect changes in kinetic energy, a channel segment with a length $l$, a channel slope $S$, and a discharge $q$ (volume per time) dissipates a power

$$P = \rho g q l S, \tag{10}$$

where $\rho g$ is the specific weight of water. Since the mean discharge is proportional to the catchment size under uniform precipitation, the mean dissipation is

$$\overline{P} \propto AS, \tag{11}$$

and in combination with Hack's relation (Eq. 2)

$$\overline{P} \propto A^{1-\theta}. \tag{12}$$

So the increase in dissipated power with catchment size is weaker than linear as long as $\theta > 0$. Then a single channel with a catchment size $A$ is energetically favorable (dissipates less energy) over two channels with $\frac{A}{2}$ each. This is the main reason why the concept of OCNs predicts dendritic networks instead of parallel channels. In turn, parallel flow patterns are energetically favorable for $\theta \leq 0$. This also includes the limiting case $\theta = 0$. Since the dissipated power is directly proportional to the 200 catchment size then, the shortest path to the boundary yields minimum energy dissipation.

    The fluvial erosion model in its original form, e.g., the SPIM (Eq. 1) or the shared stream-power model (Eq. 3) should only be applied to channelized sites according to the criterion defined in Sect. 2. In turn, we need a model for hillslopes that does not favor dendritic networks over parallel flow patterns energetically. Then there is a chance that parts of the domain do not self-organize towards dendritic channel networks, but towards parallel flow patterns. Otherwise, we should expect that the 205 entire area will be captured by channel networks, and that hillslopes will be limited to sites with catchment sizes of only a few pixels, as found for the entirely fluvial topography in Sect. 2.

Following these considerations, we need a model for the hillslopes that predicts a concavity index $\theta \leq 0$ in equilibrium. Any version of the shared-stream power model (Eq. 3) with $m \leq 0$ and $n > 0$ satisfies this condition since $\theta = \frac{m}{n} \leq 0$ in equilibrium. While $m < 0$ results in convex equilibrium profiles, $m = 0$ generates straight slopes. The shared stream-power model with $m = 0$ can be interpreted in the way that the ability to erode is independent of the discharge and that the transport capacity (Eq. 4) is proportional to the discharge.

The choice $m = 0$ is appealing since it circumvents the problem that the catchment size $A$ (or the discharge) is not suitable for describing unchannelized flow due to its dependence on grid spacing. For $m < 0$, we would need a model written in terms of catchment size per unit width or discharge per unit width as proposed by Bonetti et al. (2018). In turn, the term $\frac{Q}{A}$ at the left-hand side of Eq. (3) does not cause any problems because considering both the sediment flux $Q$ and the catchment size $A$ per unit width would not change their ratio.

## 4 A numerical test

In this section, we test the criterion for delineating channels proposed in Sect. 2 in combination with the linear version of the shared stream-power model ($n = 1$). Let us assume that hillslopes are also described by the shared stream-power model (Eq. 3) with $m = 0$ and erodibilities $\tilde{K}_d$ and $\tilde{K}_t$. For simplicity, we assume

$$\frac{\tilde{K}_d}{K_d} = \frac{\tilde{K}_t}{K_t} \tag{13}$$

and define

$$A_h = \left(\frac{\tilde{K}_d}{K_d}\right)^{\frac{1}{m}} = \left(\frac{\tilde{K}_t}{K_t}\right)^{\frac{1}{m}}. \tag{14}$$

Then the hillslopes are described by the same equation as the rivers (Eq. 3) even with the same values of $m$, $K_d$, and $K_t$, but with $A_h$ instead of $A$ at the right-hand side. Using $A_h$ instead of $\tilde{K}_d$ and $\tilde{K}_t$ will facilitate the interpretation of the results.

Furthermore, the parameter $A_h$ can be interpreted directly in terms of the efficiency of erosion at hillslopes compared to erosion in channels. It is easily recognized that $A_h$ defines the catchment size above which the erosion by channelized flow is stronger than erosion at hillslopes at the same channel slope $S$. In this sense, $A_h$ could also be defined for other models than the shared stream-power model used here. In each case, however, we should keep in mind that $A_h$ is a process-related parameter and not an imposed threshold catchment size.

The results shown in the following were obtained with the parameter combination $K_d = K_t = 2$, which can be seen as the middle between the detachment-limited model and the transport-limited model with an effective erodibility $K = 1$ (Eq. 7). However, additional simulations with the detachment-limited model and the transport-limited model revealed that none of the results rely on this choice.

Since the catchment size has no effect on erosion for $m = 0$, the choice of the flow routing scheme for hillslopes is not crucial. However, it is important that the same scheme is applied to sediment fluxes and catchment sizes in order to keep the ratio $\frac{Q}{A}$ occurring in Eq. (3) consistent. Adopting the D8 scheme from the channelized sites simplifies the implementation and

has the advantage that the fully implicit scheme proposed by Hergarten (2020b) can be used. So we apply the D8 scheme to all sites. Although it is in general not well-suited for hillslopes, we will see in Sect. 6 that it works quite well for the model considered here.

Simulations were performed for $A_h = 10$, 100, and 1000, starting from the fluvial equilibrium topography shown in Fig. 3. The simulations were run with a time increment $\delta t = 10^{-3}$. A steady state in the strict sense was not achieved in any of the simulations. A considerable number of changes in flow direction (at about 2 % of all grid cells) occurs in each time step. However, these changes mainly affect the hillslopes, while changes in channels and transitions between channels and hillslopes are rare. We will return to this aspect later in this section. The results presented in the following were derived from the topography at a large time $t = 100$ in order to ensure that there is no systematic change in topography anymore.

Figure 4 shows the parts of the obtained topography defined by the rectangle in Fig. 3. It is immediately recognized that the topography becomes smoother with increasing $A_h$. The profiles drawn in Fig. 5 confirm that the flanks of the valleys turn from almost vertical walls into straight hillslopes. The steepest segments of the profiles are as steep as expected according to Eq. (8) with $A_h$ instead of $A$,

$$S = \left( \frac{E}{K A_h^m} \right)^{\frac{1}{n}} = A_h^{-0.5}, \tag{15}$$

for $m = 0.5$, $n = 1$, $K = 1$, and $E = U = 1$. Less steep segments are an effect of the orientation of the hillslopes relative to the profile. For $A_h = 1000$, the largest river is slightly lower than for the other topographies. However, this does not mean that it is less steep. We found that the channel slopes of all rivers satisfy the expected equilibrium relation (Eq. 8) except for some small deviations owing to the dynamic reorganization. However, increasing $A_h$ does not only smoothen the topography, but also makes rivers less convoluted. As a consequence, the flow length towards the boundary decreases slightly, which explains the lower elevation.

Figure 6 shows the flow pattern of the region defined by the rectangle in Fig. 4c ($A_h = 100$). About 60 % of the area belongs to a small catchment with $A \approx 5000$. The smallest catchment size among the channels shown here is $A = 189 \approx 2 A_h$. In turn, the vast majority of the hillslope sites has a catchment size considerably below $A_h = 100$. While the catchment size of hillslope sites has no immediate meaning in the model considered here, it is relevant for the effect of potential disturbances. If a hillslope site incises, its number of lower neighbors may decrease, so that it may turn into a channel site. If $A < A_h$, however, its erosion rate will decrease then since erosion in channels is less efficient than at hillslopes for $A < A_h$, which counteracts incision. So it will likely be converted back into a hillslope site. In our numerical simulations, we found that practically all newly formed channel sites with $A < A_h$ fall back to hillslope sites rapidly – often immediately in the next step.

However, there are also hillslope sites with $A > A_h$. If such a site turns into a channel site, its erosion rate increases, which supports further incision. So hillslope sites with $A > A_h$ may turn into stable channel sites. However, Fig. 6 reveals that planar hillslopes with a parallel flow pattern are too short to reach the required catchment size. Hillslope sites with $A > A_h$ are only found where convergent flow occurs. These are predominantly regions above channel heads and above outer bends of existing channels.

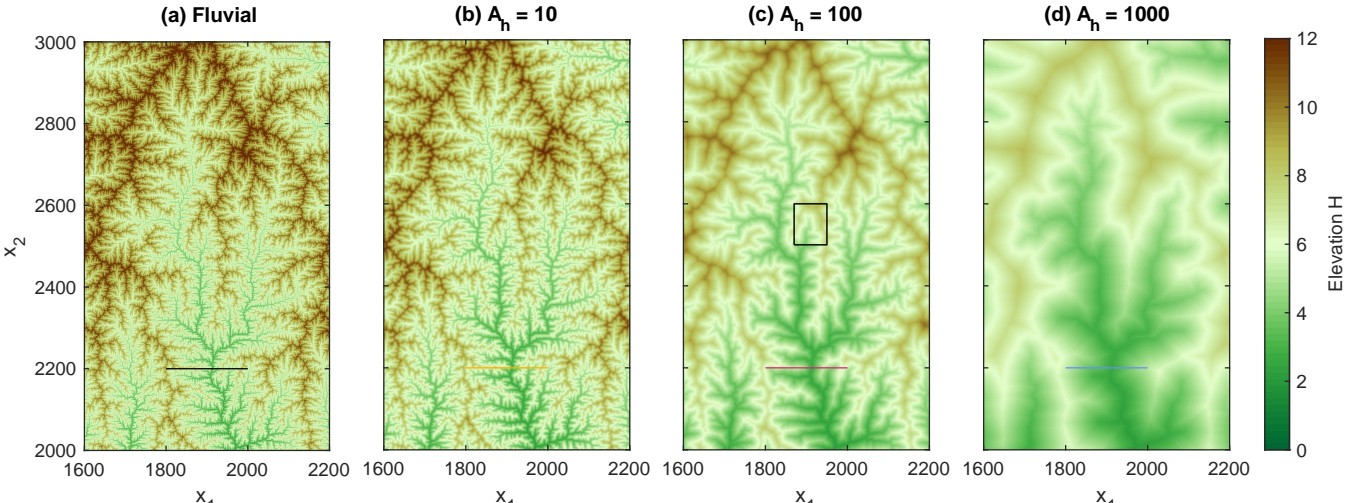

**Figure 4.** Part of the topography defined by the rectangle in Fig. 3 for different values of $A_h$ (the catchment size above which erosion in channels becomes more efficient than at hillslopes). The profile lines refer to Fig. 5 and the rectangle to Fig. 6.

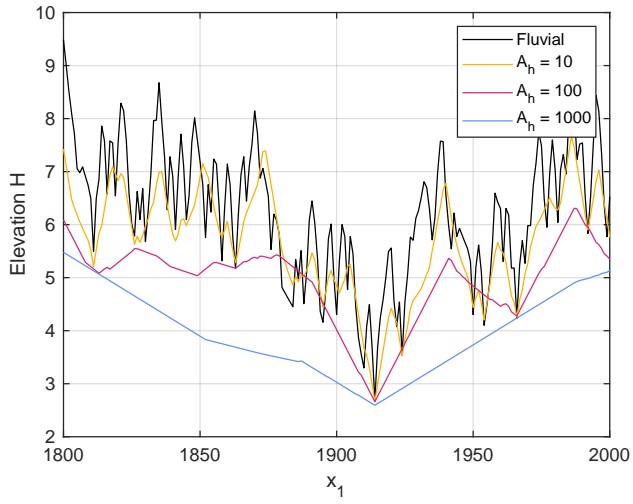

**Figure 5.** Topographic profiles along the lines defined in Fig. 4.

The respective topography is shown in Fig. 7. Hillslopes with a parallel flow pattern in Fig. 6 correspond to planar, faceted areas. While the straight longitudinal profiles are directly related to the model used for hillslope erosion ($m = 0$), the occurrence of planar patches is owing to the D8 scheme. As this scheme is not only used for computing the flow pattern (which is not immediately relevant at hillslopes), but also for computing the slope gradient, it enforces the formation of facets aligned either parallel to the coordinate axes or at a $45°$ angle. This restriction is also responsible for the large number of changes in flow direction that persist even in an almost steady state. These changes mainly affect edges between planar facets and domains

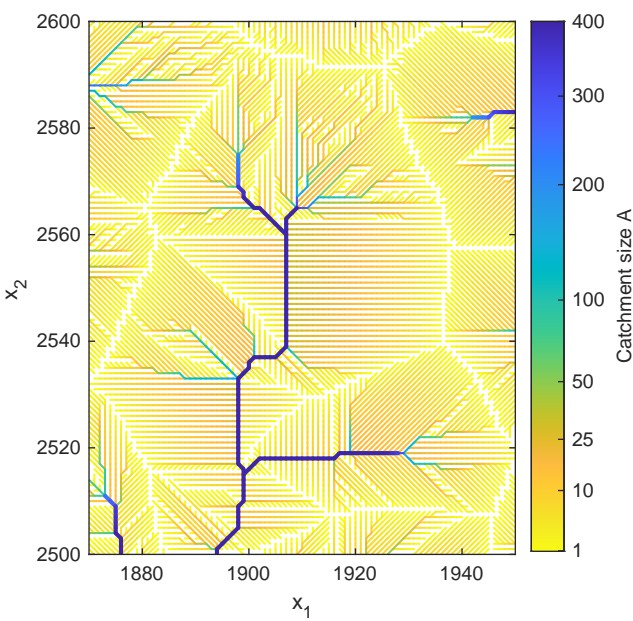

**Figure 6.** Drainage pattern of the the region defined by the rectangle in Fig. 5c. Channels are marked by thick lines. Hillslopes draining into straight river segments are typically characterized by a parallel flow pattern with catchment sizes considerably below $A_h = 100$. Catchment sizes $A \gtrsim A_h$ occur preferably at convergent hillslopes above channel heads.

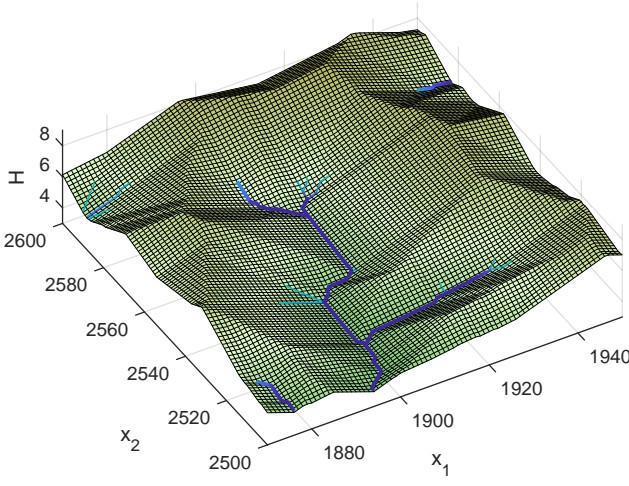

**Figure 7.** Topography of the domain shown in Fig. 6. Computing slope gradients based on the D8 scheme generates faceted, planar hillslopes.

where the large-scale orientation is not compatible with any of the 8 available directions (e.g., the upper left corner in Fig. 6). It could be said that such sites attempt to overcome the limitation in flow direction on average by changing their flow direction frequently.

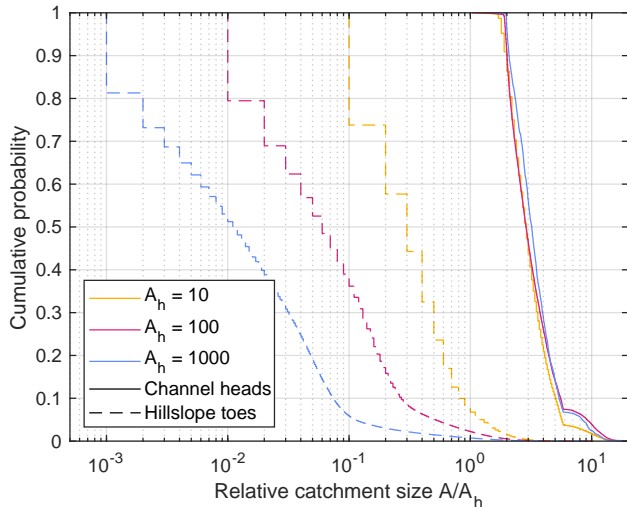

**Figure 8.** Empirical cumulative distributions of the catchment sizes of channel heads (channel forming areas, solid lines) and hillslope toes (dashed lines).

A more detailed analysis of the catchment sizes over the entire domain is given in Fig. 8. The solid lines show the empirical cumulative distribution of the channel heads, so of the channel forming areas. When rescaled to $A_h$, the distributions of the channel forming areas collapse well for the considered values of $A_h$. So the channel forming areas scale consistently with the process-based parameter $A_h$, which was not the case for the diffusion model considered in Fig. 2 in terms of the diffusivity.

As a striking property, the vast majority of all channel heads is in the range from $2A_h$ to $6A_h$. So channelization does typically not take place at the catchment size $A_h$ at which erosion in channels becomes stronger than on hillslopes, but a considerably larger catchment sizes. This property will be addressed in Sect. 6.

The dashed lines in Fig. 8 show the respective distribution for the hillslopes. For clarity, not all hillslope sites are analyzed, but only hillslope toes (hillslope sites that drain directly into a channel). It is immediately recognized that the catchment sizes at the hillslopes do not scale linearly with $A_h$. Owing to the dominance of parallel flow patterns at hillslopes, the catchment sizes at the toes rather scale linearly with the length of the hillslopes than with $A_h$. In our example, the different scaling of catchment sizes in channels and at hillslopes is not a problem since the catchment size is not relevant for the hillslopes. Otherwise, however, the results would be dependent on the spatial resolution, which should be avoided by referring to catchment size per unit width at hillslopes.

## 5   Scaling behavior

As discussed in Sect. 1 and 2, a dependence of the numerical results on the spatial resolution is an issue in many coupled models of fluvial erosion and hillslope processes. The linear increase in channel forming area with $A_h$ found in the previous section already suggests that our approach avoids such problems. However, a more thorough analysis should also involve the

topographies obtained from simulations on lattices with different resolutions, but with the same model parameters. In principle, the simulations performed in the previous section on a grid with unit spacing can be rescaled accordingly.

Let us assign a value $\delta x$ (in meters) to the unit grid spacing, a vertical length scale $L$ to one nondimensional elevation unit, and a time scale $T$ to one unit of nondimensional time. It is easily recognized from a dimensional analysis of Eq. (1) that the nondimensional erodibility $K$ has to be rescaled by a factor

$$\alpha = \delta x^{n-2m} L^{1-n} T^{-1}. \tag{16}$$

Accordingly, the nondimensional uplift rate $U$ must be rescaled by a factor $\beta = LT^{-1}$. So transferring the results of a nondi-
305 mensional simulation with unit grid spacing to scenarios with a various values $\delta x$ at constant $K$ and $U$ requires that $\alpha$ and $\beta$ are constant, and thus

$$T \propto L \propto \delta x^{\frac{n-2m}{n}}. \tag{17}$$

For the combination $\frac{m}{n} = 0.5$ used here, this even implies that $L$ and $T$ are independent of $\delta x$. These results also hold for the the shared stream-power model (Eq. 3).

However, this scaling behavior is lost if a model for hillslope processes is included. For the model considered here, $\tilde{K}_{\mathrm{d}}$ and $\tilde{K}_{\mathrm{t}}$ scale differently from $K_{\mathrm{d}}$ and $K_{\mathrm{t}}$ (Eq. 16 with $m = 0$). This different scaling introduces a characteristic horizontal length scale. In terms of the parameter $A_{\mathrm{h}}$ defined in Eq. (14), this means that the real-world value of $A_{\mathrm{h}}$ scales with $\delta x^2$. In turn, keeping all erodibilities (and thus the real-world value of $A_{\mathrm{h}}$) constant requires a scaling of the nondimensional value of $A_{\mathrm{h}}$ according to $A_{\mathrm{h}} \propto \delta x^{-2}$.

So our nondimensional simulations with $A_{\mathrm{h}} = 10$, 100, and 1000 can be interpreted as simulations with identical parameters, but different grid spacings $\delta x$ and thus also different domain sizes. For comparing the results, the relief of all catchments is shown in Figure 9. The number of catchments ranges from 1237 for $A_{\mathrm{h}} = 1000$ to 157,339 for $A_{\mathrm{h}} = 10$. Since $A_{\mathrm{h}} \propto \delta x^{-2}$, the ratio $\frac{A}{A_{\mathrm{h}}}$ on the x-axis is proportional to the real-world catchment size.

     Despite the scatter in the data, it is recognized that the relief increases logarithmically with the ratio $\frac{A}{A_{\mathrm{h}}}$ and that the data
collapse quite well for different values of $A_{\mathrm{h}}$ as expected for $\frac{m}{n} = 0.5$. Fitting logarithmic functions confirms this finding. In particular, the functions obtained for $A_{\mathrm{h}} = 100$ and $A_{\mathrm{h}} = 1000$ are very close to each other and suggest the relation

$$\delta H = 2\log_{10} \frac{A}{A_{\mathrm{h}}} + 1.75. \tag{18}$$

     Additional tests performed for $\frac{m}{n} = 0.25$ and for $\frac{m}{n} = 0.75$ did not reveal any scaling issues. It just has to be taken into account that Eq. (17) also requires a rescaling of the relief $\delta H$ with $\delta x$ for $\frac{m}{n} \neq 0.5$. However, the simple logarithmic increase
of relief with catchment size (Eq. 18) only holds for $\frac{m}{n} = 0.5$.

     Beyond this, our findings only suggest that steady-state topographies are robust against the spatial resolution. For time-dependent scenarios, the effect of the resolution should be investigated more thoroughly. As an example, Hergarten (2021) investigated properties of mobile knickpoints in the purely fluvial version of the shared stream-power model. While it was found that the speed of knickpoint migration is independent of the spatial resolution, the respective response of the sediment

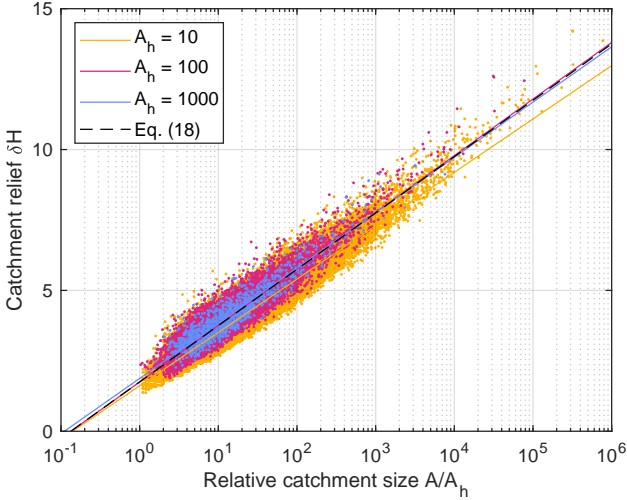

**Figure 9.** Relief of all catchments. The solid lines show fitted logarithmic functions, and the black dashed line corresponds to Eq. (18).

flux is not. This dependence was attributed to the size of the smallest (single-pixel) catchments. We would expect that our
approach removes this dependence, but this would have to be investigated in detail.

## 6   The break in slope

In the previous sections, we found that the concept for delineating channels developed in Sect. 2 works well in combination
with a simple model for erosion at hillslopes and shows a reasonable scaling behavior. We now approach the question to what
extent these results rely on the specific model and which parts can be generalized.

As the most striking result, we found a shift in catchment sizes. While erosion in channels is stronger than at hillslopes
for $A > A_h$, almost all channel heads have catchment sizes $A > 2A_h$. The following geometrical considerations show that this
result is not specific to the considered model, but to the regular lattice with the D8 flow routing scheme.

Three channel segments with different channel slopes are sketched in Fig. 10. If we apply the D8 scheme also to the
hillslopes, the surrounding hillslopes are oriented perpendicular to the channel segment as long as the channel slope is quite
low (Fig. 10a). For steeper channels, the D8 flow direction switches to the diagonal neighbor (Fig. 10b). Above a critical
channel slope, sites in the valley have more than one lower neighbor, so that the channel no longer satisfies the criterion for
channelization (Fig. 10c).

Figure 11 shows all possible scenarios for straight channel segments in plan view. For simplicity, unit grid spacing and unit
slope at hillslopes are assumed. Let us start from an axis-parallel channel segment as illustrated in Fig. 11a, corresponding to
Fig. 10. The elevations along the channel are $0, S, 2S, \ldots$, where $S$ is the channel slope. If we apply the D8 scheme also to the
hillslopes and assume that the channel is rather steep, the elevation of the red sites is $\sqrt{2}$ since they drain in diagonal direction

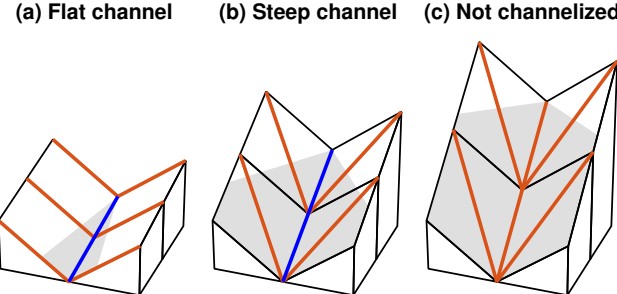

**Figure 10.** Three channel segments with different channel slopes. Blue lines refer to the flow directions of channelized flow. Red lines describe flow directions that do not satisfy the criterion for channelization. The area below the uppermost point of the channel is shaded.

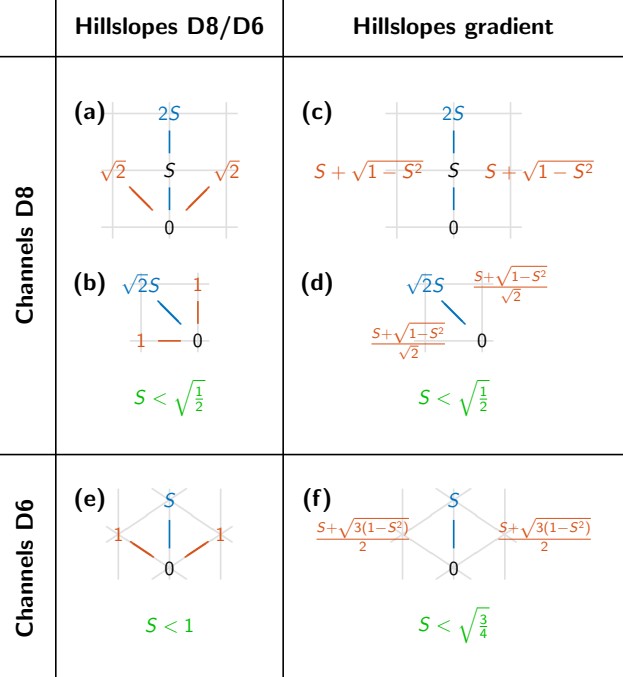

**Figure 11.** Geometry of channels and hillslopes for different topologies in plan view. Blue lines refer to flow in channels and red lines to flow on hillslopes if a single-flow-direction scheme is also used for hillslopes. Blue numbers are elevations of channel sites that must be lower than the elevations of the hillslope sites given by red numbers. Unit grid spacing is assumed, $S$ is the channel slope, and the gradient of hillslopes is unity.

to a site with zero elevation. Then the blue site can only be channelized if its elevation is lower than those of the red sites, so $2S < \sqrt{2}$.

For a diagonal channel segment (Fig. 11b), the elevation of the blue channel site must be $\sqrt{2}S$, while the elevation of the red hillslope sites is one due to their axis-parallel flow direction. So the condition for the channelization of the blue site is $\sqrt{2}S < 1$.

In both cases, the condition for channelization is $S < \sqrt{\frac{1}{2}}$. So slopes in channels must be at least by a factor of $\sqrt{\frac{1}{2}}$ lower than at the surrounding hillslopes. In order to achieve the same erosion rate, the catchment size must be

$$355 \quad A = \sqrt{2}^{\frac{n}{m}} A_{\text{h}} = 2^{\frac{1}{2\theta}} A_{\text{h}} = 2A_{\text{h}} \tag{19}$$

for $\theta = 0.5$ in the channel according to Eqs. (8) and (15). So the finding $A > 2A_{\text{h}}$ relies on the D8 scheme and on our choice $\theta = 0.5$.

The break in slope by a factor of $\sqrt{\frac{1}{2}}$ for all orientations (axis-parallel or diagonal) of the channel segment is crucial for the applicability of the D8 scheme. If the factors were different, we would expect problems with anisotropy. Then either
axis-parallel or diagonal channel segments would be preferred in the upper ranges of rivers in combination with a preferred orientation of the surrounding hillslopes.

Using a representation of the gradient by difference quotients at hillslopes does even not affect the factor $\sqrt{\frac{1}{2}}$ (Fig. 11c,d). In order to obtain a total slope of one at a given channel slope $S$, the slope perpendicular to the channel must be $\sqrt{1-S^2}$. This yields an elevation of $S + \sqrt{1-S^2}$ for the red site in Fig. 11c. For a diagonal channel segment (Fig. 11d), the respective
elevation is by a factor of $\sqrt{\frac{1}{2}}$ lower due to the shorter distances. In both cases, the obtained criterion for channelization is $S < \sqrt{\frac{1}{2}}$ and thus the same as before. So it makes no difference for the break in slope between hillslopes and channels whether we allow arbitrary slope directions at hillslopes or use the D8 scheme.

The $45°$ steps in flow direction are the reason why the simple D8 scheme performs well in combination with the criterion for channel formation. Hillslopes are perpendicular to large channels (small channel slope) and are aligned at a $45°$ angle for
the steepest possible channels. So the D8 scheme captures both end-members well. Only hillslope sites that drain directly into a diagonal channel segment are an exception since the D8 scheme only allows a $45°$ angle here. However, this is not a serious issue since it only concerns a single row of sites and does not affect the rest of the hillslopes.

Orientations between these two end-members are not captured if the simple D8 scheme is applied to the hillslopes. However, we did not encounter any obvious artifacts that could be related to this limitation. So the simple D8 scheme appears to be well-
suited not only for the channels, but also for the hillslopes. This is an advantage for the numerical implementation since it allows for a seamless application of the fully implicit scheme proposed by Hergarten (2020b).

An isometric grid consisting of equilateral triangles may provide a better isotropy than a regular grid at first sight. If the gradient is used for the hillslopes, the slope break between hillslopes and channels is smaller than for the D8 scheme, owing to the lower number of competing neighbors (6 instead of 8). As illustrated in Fig. 11f, the respective factor is $\sqrt{\frac{3}{4}}$ instead
of $\sqrt{\frac{1}{2}}$. More important, the slope break vanishes completely if we use the slope towards the lowest neighbor (called D6 in Fig. 11) at hillslopes (Fig. 11e). Furthermore, the restriction to the lowest neighbor aligns all hillslopes at an angle of $60°$ towards the respective channels, so that the gradient of hillslopes draining into large channels (with low channel slopes) would

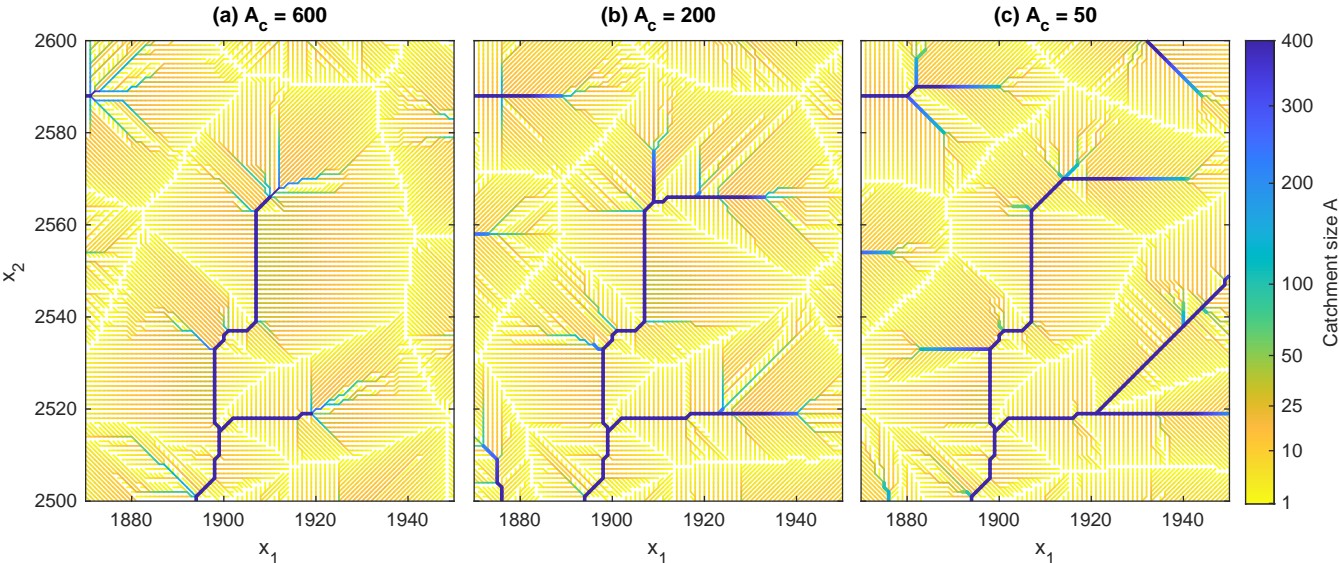

**Figure 12.** Drainage pattern of the the region shown in Fig. 6 for $A_\mathrm{h} = 100$ and different values of the channelization threshold $A_\mathrm{c}$. Channels are marked by thick lines.

not be captured well. While we did not perform any numerical tests on triangular grids, these results suggest that regular grids in combination with the D8 scheme are better in this context, owing to the $45°$ steps in direction.

## 7 Defining channel thresholds explicitly

In its spirit, the idea of distinguishing channels from hillslopes by the topography differs from the more conventional concept based on a pre-defined threshold catchment size $A_\mathrm{c}$ for channelization. As a major difference, the topography-based approach does not enforce a strict threshold for the initiation of channels. For the model investigated in Sect. 4, most of the channel heads are in the range $2A_\mathrm{h} \leq A \leq 6A_\mathrm{h}$. Beyond this variation by a factor of three, $A_\mathrm{h}$ is not an additional model parameter, but was derived from the parameters of the erosion models (Eq. 14). It describes the catchment size at which channel erosion becomes more efficient than hillslope erosion.

In order to find out to what extent both approaches differ practically, we performed simulations with the same model, but with an explicit threshold $A_\mathrm{c}$ for channelization instead of the criterion based on the number of lower neighbors. Figure 12 shows the drainage pattern of the region from Fig. 6 for different values of $A_\mathrm{c}$. All model parameters are the same as in Sect. 4, including $A_\mathrm{h} = 100$. The threshold value $A_\mathrm{c} = 600$ (Fig. 12a) then corresponds to the maximum catchment size of more than 90 % of all channel heads in the self-organizing model (Fig. 8). In turn, $A_\mathrm{c} = 200$ (Fig. 12b) corresponds to the minimum catchment size obeyed by almost all channel sites in the self-organizing model. For $A_\mathrm{c} = 50$ (Fig. 12c), channel segments may be steeper than the surrounding hillslopes, which makes channels with $A < 100$ unstable.

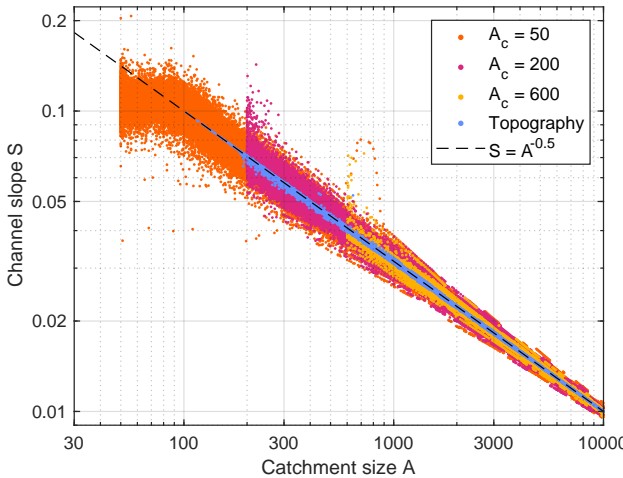

**Figure 13.** Channel slopes of all channelized sites for $A_\mathrm{h} = 100$ and different values of the channelization threshold $A_\mathrm{c}$. The blue dots refer to the topography-based criterion without threshold. The dashed line shows the theoretical equilibrium relation $S = A^{-0.5}$ (Eq. 8).

The pattern of the largest rivers (which are still rather small) is identical to that from Fig. 6. Measured over the entire topography, only about 3 to 5 % of all sites with $A \geq 1000$ change their flow direction compared to the self-organizing model without threshold. As expected, the channels extend more into the hillslopes with decreasing $A_\mathrm{c}$. As a consequence, the upper parts of the channels tend to be unstable, which leads to an increased frequency of reorganization.

This effect is immediately recognized in the analysis of the channel slopes shown in Fig. 13. While the equilibrium channel slope is $S = A^{-0.5}$ according to Eq. (8), a considerable scatter is found in the actual channel slopes. This scatter decreases with increasing $A_\mathrm{c}$, but is stronger than for the topography-based criterion for all considered values of $A_\mathrm{c}$. Accordingly, there is a strong variation in erosion rates, which indicates a rapid reorganization of the drainage pattern at small catchment sizes. The resulting fluctuations in sediment flux are responsible for the downstream propagation of the scatter, which is still visible at $A = 10000$.

A distinct change in channel slopes occurs at $A = A_\mathrm{h}$ (which requires $A_\mathrm{c} < A_\mathrm{h}$). The systematic decrease in $S$ with $A$ is even lost for $A < A_\mathrm{h}$. This is the situation where equilibrium channels would be steeper than hillslopes and thus cannot be stable. Then the headwaters are formally channels ($A > A_\mathrm{c}$), but rather hillslopes in their properties. So the erosion law (the efficiency of hillslope erosion compared to fluvial erosion, expressed by $A_\mathrm{h}$ here) overrides the threshold of channelization in this case.

These results suggest that the model somehow counteracts the imposed threshold $A_\mathrm{c}$ by permanently switching between channels and hillslopes for all considered values of $A = A_\mathrm{c}$. It looks as if this model was constrained too strongly. In each case, using different models for channels and hillslopes introduces a characteristic catchment size $A_\mathrm{h}$ above which channels erode more efficiently than hillslopes. Defining a second characteristic catchment size by imposing a threshold at which hillslopes turn into channels seems to be a condition too many.

However, a permanent reorganization of the drainage pattern is not unusual in fluvial erosion models and is not necessarily
a problem. It arises from an interplay of small changes in the flow pattern and in elevation, which propagate upstream towards
the drainage divides and may therefore cause ongoing oscillations. The susceptibility of the model to such oscillations depends
on the channel slope at drainage divides since steeper drainage divides can accommodate larger changes in elevation without
changing the discrete flow pattern. Since hillslope processes make drainage divides less steep, models that include hillslope
processes typically do not achieve a steady state. This also holds for the examples with diffusion considered in Sect. 2. The
large number of changes in flow direction at the edges of faceted hillslope segments observed in Sect. 4 arises from a different
mechanism, but is also related to the discrete flow pattern.

The switches between channels and hillslopes found in this section are different from the oscillations described above since
they are not restricted to individual sites that change their flow direction. While such a formation of temporary channels on
hillslopes is not necessarily unrealistic, it may also make the model more complicated from a theoretical point of view. Since
catchment size is not well-defined on hillslopes, it may generate artifacts. However, analyzing the relief the same way as in
Fig. 9 did not reveal any obvious scaling issues. So we cannot pinpoint any clear problem of the concept based on a threshold
catchment size for the transition to channelized flow at this stage.

## 8   Perspectives

Finally, the question arises how to proceed concerning the potential scaling issues in coupled fluvial–hillslope landform evo-
lution models. Avoiding the application of fluvial erosion models that were developed for channels to parallel flow patterns at
hillslopes seems to be the key to solving the scaling issues.

This question is in principle independent of the model used for the hillslopes. While we used an extension of the shared
stream-power model towards hillslopes for illustration, the arguments would be basically the same for the more widely used
diffusion models. This also includes the nonlinear diffusion model introduced by Roering et al. (1999), which enforces an
upper limit for the slope and is nowadays widely used. Combinations would also be possible, such as adding diffusion to the
extended shared stream-power model at all sites. Theoretically, our concept of self-organization only requires that the model
used for hillslopes generates convex ($\theta < 0$) or straight ($\theta = 0$) topographies. All models discussed here satisfy this condition.

Concerning models that apply fluvial and hillslope processes to the entire domain, there is no immediate reason why replac-
ing diffusion by any other model should solve the scaling issue discussed in Sect. 1 and 2. So we would at least have to be
aware of potential scaling issues in such models and to be careful concerning the spatial resolution.

The topography-based self-organization proposed in this study as well as threshold-based models seem to be quite robust
against scaling issues. Threshold-based models, however, suffer from the problem that the threshold typically depends on the
involved processes in nature. Channel initiation is not a only a matter of the fluvial processes, but also depends on hillslope
processes that may counteract incision. Coupled models already include this information implicitly, and our results on self-
organization suggest that they attempt to adjust accordingly. Formally, this means that each combination of models contains

a catchment size $A_h$ (which is not necessarily constant) above which channels erode more efficiently than hillslopes, and this catchment size controls the formation of channels.

In the previous section, we saw that defining a threshold catchment size $A_c$ counteracts the self-organization and causes a battle of two competing scales. We also recognized that this battle results in strong oscillations, but not necessarily in scaling issues. However, we would have to be check whether this is still the case for the considered combination of models if the two scales differ strongly. In particular, we have to be careful with incision thresholds. This concept dates back to Montgomery and Dietrich (1992) and assumes that channel initiation is not only dependent on catchment size $A$, but also on channel slope $S$, where typically the same combination is used as at the right-hand side of Eq. (1). Making the topography steeper (e.g., by increasing the uplift rate) extends the channels towards smaller catchment sizes. Theoretically, we could even enforce a channelization of the entire domain, which would likely cause scaling issues in combination with hillslope processes.

Our results suggest that the self-organization of channels and hillslopes based on the topography is a simple and robust approach to circumvent all these issues. As part of its simplicity, it contains no additional parameters. The property $A_h$ used for analyzing the results is not an independent parameter, but derived from the parameters of the erosion models. In turn, however, an ad-hoc model for delineating channels was used. This model is reasonable, but not unique. While the minimum channel forming area of stable channels is proportional to the process-related property $A_h$, the factor of proportionality relies on the model for delineating channels and on the D8 flow routing scheme. Achieving the same minimum channel forming area on a grid with a different topology would require a modification of the scheme for delineating channels, which may cost a part of the simplicity. In total, however, all these potential complications seem to be minor compared to the challenge of determining a threshold $A_c$ that is compatible with the parameters of the erosion models manually.

## 9   Conclusions

In this study, a new concept for coupling fluvial erosion and sediment transport with hillslope processes in landform evolution models is proposed. In contrast to the more conventional approaches based on a pre-defined threshold catchment size for channelized flow or an incision threshold, this concept directly uses the topography and aims at a self-organization of channels and hillslopes. Channelized flow is assumed for all sites of a discrete grid that have only one neighbor with a lower elevation. This definition reflects the idea that a thin layer of water is focused into a single direction without spreading laterally. Theoretical considerations based on energy dissipation suggest that it depends on the model used for erosion at hillslopes whether a self-organization of channels and hillslopes is possible. In general, all models that predict convex or straight equilibrium topographies at hillslopes should be suitable.

In order to test the concept numerically, we combined the shared stream-power model for fluvial erosion with a simple model for hillslopes, where the erosion rate only depends on slope. As a main result, the topography indeed self-organizes into channels and hillslopes. Channel heads form in a certain range of catchment sizes. This range depends on the parameters of the models used for channels and hillslopes. The dependence can be expressed in terms of the catchment size at which erosion in channels is more efficient than at hillslopes ($A_h$ in the formulation used here), but is considerably higher. So the actual

transition from hillslopes to channels takes place at larger catchment sizes where erosion in channels is substantially more efficient than at hillslopes. This effect can be explained by a gap in slopes between channels and hillslopes. For the simple D8 flow routing scheme, channels must be at least by a factor of $\sqrt{\frac{1}{2}}$ less steep than hillslopes.

The numerical tests revealed no obvious dependence of the results on the spatial resolution, which is a typical problem in coupled models in which fluvial erosion and hillslope processes act on the entire domain. The approach works well even if the D8 scheme is used for computing gradients at hillslopes. While this simplification allows for a seamless coupling of fluvial erosion and hillslope processes, it enforces the formation of faceted areas on hillslopes in combination with a permanent reorganization. However, the effects of this reorganization on the large-scale topography seem to be minor.

Finally, the question arises whether the concept of self-organization based on topography proposed here is better than defining a threshold catchment size or an incision threshold explicitly. Our numerical tests revealed that the coupled model counteracts the imposition of a threshold by a strong reorganization, which also affects the channel slopes of the rivers. However, our tests did not reveal any scaling issues arising from this behavior. Nevertheless, the self-organizing model seems to be more robust than the threshold-based version.

As a second advantage, the concept based on self-organization involves no additional parameters. So we would not have to think about potential dependencies of a threshold on the parameters of the erosion models. In this sense, the self-organizing model is almost as simple as models in which fluvial and hillslope processes act on the entire domain. In turn, however, the concept has been tested so far only for a specific combination of models and for the simple D8 flow routing scheme on a regular grid. Applying the concept to other topologies may require an extension of the scheme for delineating channels. Overall, the potential influence of the simple scheme used for delineating channels has to be investigated further.

In total, delineating channels by topography and leaving the self-organization of channels and hillslopes to the respective erosion models seems to provide a simple and robust concept for coupling fluvial erosion with hillslope processes.

*Code and data availability.*  All codes are available in a Zenodo repository at https://doi.org/10.5281/zenodo.6794117 (Hergarten, 2022c). This repository also contains the data obtained from the numerical simulations. Users who are interested in using the landform evolution model OpenLEM in their own research are advised to download the most recent version from http://hergarten.at/openlem (Hergarten, 2022b). The authors are happy to assist interested readers in reproducing the results and performing subsequent research.

*Author contributions.*  S.H. developed the theoretical framework and the numerical codes. Both authors wrote the paper.

*Competing interests.*  The authors declare that there is no conflict of interest.

*Acknowledgements.* This work was funded by the Deutsche Forschungsgemeinschaft (DFG, German Research Foundation) – 432703650.

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
