# Peer review of "Correspondence: Stefan Hergarten (stefan.hergarten@geologie.uni-freiburg.de)"

_EGUsphere, 2022_

## Author Response (AR1)

Dear Reviewers, dear Editor,

thank you very much for your constructive comments!

As the main lesson from the comments, we learned that the awareness of scaling issues in landform evolution models is lower than we thought. Under this aspect, our work might even look a bit like solving a non-existing problem. We tried to point out more clearly that these scaling problems exist and that it makes sense to solve them (see below).

The points addressed in the four reports are discussed below, where changes to the manuscript are highlighted in bold letters. Line numbers refer to the version with highlighted changes.

Best regards,

Stefan Hergarten and Alexa Pietrek

**Reviewer 1 (Alan Howard)**

*The detailed exposition of the modeling scheme and the results is generally clear. Most comments concern basic questions about the model implantation and definition of channels.*

*General comments*

*Section 2 general comment: The erosion model in this part of the paper only considers fluvial processes. In the absence of slope processes is not the entire network channelized by definition?*

Concerning the model considered for illustration in this section, you are right that the reference scenario (solid black line in the new Fig. 2) is purely fluvial. The result is that the simple scheme does not recognize channelization up to the drainage divides (single-pixel catchments), but misses only small catchments. Maybe this becomes clearer in the revised version, although it should also have been clear in the first version.

*The use of $A_h$ to emulate slope processes in what is essentially a fluvial-only model seems arbitrary. The justification seems to be primarily to allow computationally-efficient modeling by eliminating the complexities of explicit modeling of mass wasting. This lessens any general implications of scale independence beyond their specific LEM.*

Of course, we are not able to prove formally that our simple scheme for delineating channels is free of scaling problems for any combination of fluvial/hillslope models. So we have to use an example for illustration (Sect. 4 and 5). In some sense, this example is the simplest realization of the theoretical concept presented in Sect. 3 – straight hillslopes from a hillslope erosion rate that only depends on the slope and a transport capacity that is proportional to the catchment size. Anyway, you may say that it is somehow arbitrary.

*Specific comments*

*The issue of defining the channel network and, correspondingly, drainage density is not adequately discussed. In natural drainage networks one way of defining drainage density is the presence of actual channels with well-defined banks. This approach is, of course, not useful for landform evolution modeling at basin scale, at least at the level of process generalization in current LEMs. It also suffers in a more general sense that the drainage network so defined is time-dependent, because the channel network can expand and contract with flood events, land use change, and short-term climate changes that do not strongly affect drainage basin morphology as a whole. As the authors indicate, defining the channel system by a critical drainage area, $A_c$, is arbitrary. There are two ways to define $A_c$, and each has limitations. The first is straightforward imposing definition of the channel network to initiate at the critical area no matter what the modeled or actual landform processes are or what the landform morphology reveals. The second is to impose a process threshold in LEMs at a critical contributing area. That seems to be what the authors imply based upon discussion and simulations in section 7. There is some logic to this if the threshold is clearly defined as occurring at a critical process threshold, such as a critical fluvial shear stress for channel incision or a critical threshold for hillslope stability. Several studies have explored stochastic forcing of a critical fluvial shear stress (as a result of storm intensity) and its effect on drainage basin morphology. Both uses of $A_c$ are relevant only to simulation modeling and not for determining drainage density in natural networks.*

This aspect was indeed not addressed in the manuscript. We have added some discussion on this aspect at the end of the introduction, although going less deep into detail as the reviewer's comment. We hope that these paragraphs are also help to explain the motivation behind this study **(lines 145–151)**.

*The authors implement a more general scheme based upon the slope relationships between adjacent cells which is applicable to LEMs but probably error-prone for analyzing natural drainage basins. This seems to be a reasonable, but not unique approach for analysis of simulation models, although as discussed, it strongly dependent on the flow routing scheme and the use of square simulation cells. A possibly more general and less noisy method is to define a critical topographic concavity to define channel heads. Howard (1994), for example, used the gradient divergence divided by the basin-wide average gradient.*

Right, we added a short discussion on this aspect including a reference **(lines 212–217)**.

*Many LEMs utilize both fluvial and slope processes within each cell, with the emergent landscape depending upon the relevant process balance in each cell. The authors criticize this approach without providing specific justification.*

It was not our intention to criticize any model, and in particular not old models that were developed at a time when it was practically impossible to investigate scaling properties numerical. The problem is rather that we expected that the community was more aware of the scaling issues of such models. This has hopefully become clearer in the revised version **(lines 119–139 and 218–233)**.

*In actual simulations there is generally a very abrupt downgradient break between cells in which slope processes dominate and fluvial processes are unimportant and the inverse. In fact, in natural landscapes both fluvial processes and mass wasting processes occur on slopes, and individual locations can temporally transition between being dominated by either process, justifying this approach. The relative dominance of fluvial versus mass wasting process determines drainage density (Howard, 1997, EPSL, 22, 211-227; Tucker & Bras, 1998, WRR, 34, 2751-2764) and inferentially channel network definition.*

This is true, but we even believe to see a break in slope from the hillslopes to the channel heads, although we never investigated it systematically and would not claim that is by the factor $\sqrt{2}$ predicted by our criterion for channelization. It would probably be possible to replace the simple criterion by a continuous function (from 0 at hillslopes to 1 in channels, as proposed by Willgoose et al. 1991) in order to achieve a smoother transition in slope. However, the goal is to investigate the self-organization of hillslopes vs. rivers, and there it would not be very useful to blur the transition.

*The introduction should be more explicit about the implied issue in LEMs about cell-size dependency of LEM simulations. If processes are scaled correctly, there should not be cell size dependence unless the cell size is too small to adequately represent slope processes and morphology.*

Some paragraphs elaborating the issue has been added to the introduction **(lines 119–139)** as well as an additional example showing the effect of the scaling problem in Sect. 2 **(lines 218–233)**.

*Line comments*

*Line 32: If the comment about lack of sediment transport refers to the Howard (1994) model, this attribution is incorrect because sediment transport is considered as are alluvial channels. Even at locations where the channel is bedrock, the sediment is routed downstream and influences the gradient of any alluvial channel segment downstream.*

This is, of course, not the case. The reason for citing this paper here was just that we did not find an earlier reference where the term "detachment-limited erosion" was used explicitly. Anyway, we found another opportunity to cite this paper at another place **(lines 124–125)** and removed it here **(line 31)**.

*Lines 37, 202: The description of simulations such as Fig. 2 as having canyon-like morphology seems inappropriate. It seems that canyon-like seems to be conflated with the appearance of a strongly elaborated drainage network. The common usage of canyon implies a valley sharply incised into a relatively smooth upland often implying cliff-like slopes bordering the valley. I suggest using a different term and defining its meaning.*

Ok, we avoided this term in the revised version **(lines 112 and 314)**.

*Lines 61-96: The exposition here is clear. The use of combined sediment transport and bedrock erosion is fine, although in many natural channel systems transition from bedrock reaches to alluvial reaches is abrupt.*

This is true, and OpenLEM offers the option to switch to transport-limited erosion ($K_t \rightarrow \infty$ in alluvial ranges (Hergarten 2022, doi 10.5194/esurf-10-672-2022). However, discussing this aspect would go too deep into a model that is only used for illustration.

*Line 105: The choice of only one downstream cell to define channels is fine for steady state incision, but what happens when channels aggrade and there are multiple downstream potential flow paths (e.g., alluvial fans)? I see that this is addressed as a limitation in Lines 108-110.*

Distinguishing downstream bifurcations of channels from hillslopes is indeed a fundamental issue, and assuming that channels never turn into hillslopes is an ad-hoc approach. However, this issue is not specific to the approach proposed here.

**Reviewer 2**

*In this contribution, the authors take on the issue of grid scale dependence in coupled channel-hillslope landscape evolution models. Using a recently published mathematical formulation along with a new definition of what constitutes a channel, they construct an LEM that seems to circumvent some of the potential issues with past LEM implementations: chiefly the issues of 1) needing to define arbitrarily channels versus hillslopes (some previous approaches do this, though many do not) and 2) grid scale dependence in models that couple river and hillslope evolution.*

*I find the manuscript to be useful to the community given that we are always looking for modeling approaches that allow us to circumvent known issues with our current techniques. I think the authors have been honest about where the utility of their advance begins and ends (i.e., maybe not useful for running simulations on real DEMs), and that they do a nice job of exploring the behavior of the approach they propose.*

*My chief concern about the manuscript in its current form is that it is not well enough integrated into the LEM literature. Readers are not shown clearly and specifically the shortcomings of other approaches, and it is therefore hard to be sure as a reader exactly in what situations this new approach is a major advance. I would like to see this paper published in ESurf after some modifications—to the writing rather than the science—that 1) clarify the position of the current work in relation to the large body of LEM literature, and 2) clarify the utility of the current approach given its intricate relationship to the D8 grid and its current lack of applicability to non-model-generated grids. To be clear, I think this is a useful contribution, but I think the authors could increase their impact by expanding on these points.*

As a major point, we thought that the community was already more aware of the scaling issues that occur when fluvial and hillslope models are applied to the entire domain. We added some paragraphs elaborating this issue to the introduction **(lines 119–139)** as well as an additional example showing the effect of the scaling problem in Sect. 2 **(lines 218–233)**. This discussion hopefully clarifies that the problem exists and is just not immediately visible at low spatial resolutions. Since we do not propose an alternative LEM, but a concept for getting solving these scaling issues based on self-organization, the relation to the large number of existing models should also become clearer. Concerning the D8 scheme, however, we think that it should be clear to the readers that the idea itself can be transferred to other topologies in principle, but some of the results from the example rely on the D8 scheme.

*Specific comments*

*For example, the paragraph beginning on line 34 states what I believe is a well-known limitation of Eq. 2, for example Kwang and Parker 2017 talk about this. I would like to see a citation to either that work, other relevant work, or a combination of the two that demonstrates to readers that this is a known issue.*

While we agree that the integration into the overall LEM literature was a bit weak in the original version, we are not fully convinced that this paragraph is a good example. The occurrence of steep walls towards drainage divides in purely fluvial models is a direct consequence of Hack's findings and should be clear to the readers. It would neither be useful to search for the paper in which is was mentioned explicitly first nor just to cite a recent study that mentions ist. As a side note: We disagree to the conclusion by Kwang & Parker (2017) that the absence of an intrinsic horizontal length scale in a purely fluvial model for $\frac{m}{n} = 0.5$ is unrealistic. So one of us cited this study in two papers, having to express disagreement in order to justify the choice $\frac{m}{n} = 0.5$. However, we do not think that this is necessary in each paper. Anyway, we found a place in our manuscript that allows for citing this paper without reopening this aspect **(line 221–222)**.

*Similarly, in the following paragraph (line 40) there is no citation to papers describing the problem of grid scale dependence. Readers are left to wonder whether this is a problem inherent to the SPIM/diffusion model in every case, or whether a model could in theory be scaled correctly in order to avoid it. Following this discussion, in line 44, it would be good to be more specific than to say that a given approach is "not free of problems." What are the problems? Are they problems that the new method being introduced will solve, presumably?*

This points should hopefully be much clearer after elaborating the scaling issue in the introduction **(lines 119–139)** and introducing an additional example in Sect. 2 **(lines 218–233)**. We also hope that these explanations make it clearer to the readers that these problems cannot be solved completely by a rescaling and that the new concept solves exactly these problems (not only presumably).

*Another example is in the paragraph starting in line 45. We miss references to the work of Garry Willgoose (e.g., 1991 papers), who if I am not mistaken was one of the earlier workers to explicitly separate channel and hillslope process representations dynamically. I congratulate the authors on an interesting paper.*

Ok, we are also not sure whether Gary Willgoose was really the first who used a channel indicator function, but a reference to that of his 1991 papers we find most important makes sense **(line 141)**.

*Section 7 in general: I find this discussion somewhat difficult to follow, largely because I do not fully understand why the landscape is given to reorganization even under a largely steady state condition. Some more detailed explanation of the processes causing that behavior would clarify this section.*

Ok, we added some more explanation about this topic **(lines 341–345)**.

*I would also like to see a brief addition (this could also be before the conclusions if the authors prefer) making the case for why and how future workers should take advantage of the advances provided by this paper given the limitations (which are already well-stated by the authors). What can we do with this new knowledge?*

This is a good point. We added a new section (**Sect. 8**), which hopefully shows that the advantages are much bigger than the limitations.

*Line comments*

*31: I am not sure that the Howard 1994 citation is well-captured by this statement. It might be better to rephrase or to choose a reference like maybe Braun and Willett 2013 where sediment truly is not considered.*

We just wanted to give a reference to the first paper we know that used term "detachment-limited erosion". Anyway, we found another opportunity to cite this quite fundamental paper at another place **(lines 124–125)** and removed it here **(line 31)**.

*33: It would be worth pointing the reader to some of the foundational papers on modern sediment-flux-dependent river incision models, e.g. Sklar and Dietrich, 1998; Whipple and Tucker, 2002; Gasparini et al., 2007; Turowski et al., 2007, as well as the long history of models (some of which are cited later in this paper like Davy and Lague 2009) that have computed the sediment mass balance in addition to calculating river incision. We don't want readers to get the impression that we don't have options beyond detachment-limited treatments.*

We restructured the introduction in such a way that the SPIM has become less prominent, pointed out that all models described in the cited review papers contain sediment transport and added references to two more recent models **(lines 54–56)**. However, we are not doing easily with "shopping lists" of papers that have already been cited very often. For us, the immediate relevance to the topic is important, and the topic is not representation of sediment transport in fluvial landform evolution models.

*37: The use of "canyon-like" is unclear here. Do you mean landscapes in which channels become very steep at low drainage area (e.g., Kwang and Parker, 2017)? I think a new phrase is needed for clarity, or the phrase could simply be deleted and the sentence would stand as-is.*

Ok, we avoided this term in the revised version **(lines 112 and 314)**.

*41-42: I am not sure all readers will understand intuitively why this happens. Another sentence or two describing why adding hillslope processes causes a spatial resolution dependence would be useful for setting up the problem.*

This is hopefully clearer in the new explanation of the scaling issues **(lines 119–139)**.

*44: Given that the problems associated with coupled channel-hillslope models make a major motivation for this work, I ask the authors to please summarize the problems discussed in Hergarten et al. (2020a) that they reference here.*

This aspect is hopefully also clearer now **(lines 119–131)**.

*50: Again here, given that this paper is a separate contribution, it would be good to restate for readers what actually is the approach proposed by Hergarten et al (2020a).*

In the restructured explanation of the scaling issue, we get around explaining this approach in detail since it did not solve the problem completely. Now we just need the information that it is effectively the same as defining a finite river width **(lines 124–126)**.

*55: Is it possible to be more descriptive/clear than "weird?" I know that in some cases the issue with this approach is that slope-area data no longer reveal a smooth hillslope-channel transition that is observed in many real landscapes, for example. Are there other specific issues that could be brought up here? Are there citations that could be added that illustrate these issues?*

Right, this originally referred to the situation if the imposed threshold does not fit well to the process parameters. Following a suggestion of the first reviewer, we explain the relation between threshold and process parameters now in a different and hopefully clearer way **(lines 145–151)**.

*200: It is not clear why these small-scale persistent changes in the topography occur—could a sentence be added to explain more clearly?*

Unfortunately, this is not so easy at this point. So we have to put the readers off a bit here **(lines 311 and 341–345)**.

*202: Here it sounds like "canyon-like" just means "steep hillslopes." I recommend re-phrasing for clarity.*

Ok, we avoided this term in the revised version **(lines 112 and 314)**.

*209: This last sentence could use a little more detail to be clearer.*

We added a short explanation **(lines 321–323)**.

*214-215: This feels like a stupid question, but: Why does the erosion rate decrease? Is this only for the case of the chosen fluvial versus hillslope parameter values, or is this universal?*

Although it is theoretically straightforward, it is probably not a stupid question for the majority of the readers. It is directly related to the property that $A_h$ is the catchment size above which channels erode more efficiently than hillslopes. We added a note **(line 329)** and extended the explanation of $A_h$ a bit **(lines 292–296)** in order to emphasize this fundamental property of $A_h$. Now it should also be clear that this property could be transferred to any other model.

*218: Similar question for erosion rate increase. I am having trouble understanding, and I fear readers will too, what dynamics are occurring here. A few more details would help.*

Basically the same argument, just inverted. Maybe a good test for the readers after reading the previous paragraph.

*Section 5: I find this section very interesting. Do the authors expect the same result when $m/n \neq 0.5$? There are some applications in which 0.5 is a bit of a special value (Kwang and Parker, 2017) so it might be worth checking another ratio.*

Maybe a bit disappointing, but (as expected) we did not find any exciting differences. The respective figures for $\theta = 0.25$ and $\theta = 0.75$ are shown on the following page. The respective analyses of Fig. 8 are also consistent with the theoretical prediction (from the slope break). So $\theta \neq 0.5$ is just more complicated, but there is no fundamental difference. Anyway, it was worth testing it in order to be sure. We added a few sentences **(lines 395–397)**.

[Figure]

[Figure]

*Same analysis as in Fig. 9 for $\theta = 0.25$*

[Figure]

Same analysis as in Fig. 9 for $\theta = 0.75$

*272: Like many of the literature references throughout, this one is quite vague. Could the authors add an extra sentence clarifying what salient points of that paper are relevant? I for example am aware of Hergarten 2021 but have not read it in any detail, so am a bit lost here.*

Ok, we added a few sentences **(lines 401–404)**, which hopefully give some idea what is is about.

*328: Again it would be good to see multiple references here to demonstrate the extent to which this practice is established. Certainly this is an assumption in much topographic analysis of real DEMs, but in my understanding of the literature it has not (at least recently) been a favored approach for LEMs. If I am wrong, then thats ok and the addition of several citations will settle the question.*

Ok, "established" should not mean that the majority uses this approach. We replaced it by "more conventional" **(lines 460 and 546)**.

**Reviewer 3**

*This is an interesting manuscript which introduces a new idea of implementing landscape evoution simulations. I have found that many of my technical concerns are already commented by other referees.. and so try to add comments which were not mentioned yet.*

*My major concern is that the focus of the manuscript is somewhat distracting. I understand the value of new modeling framework, but I am uncertain how this can lead to any new findings or scientific advances in self-organization processes. In particular, the OCN contents in section 3 are not well harmonized with the rest of the manuscript. I suggest in the revision that authors decide the focus of this manuscript sharply, and restructure the writing.*

Sorry, but the second part is the key to the concept proposed here, and the OCN concept is the basis. We tried to integrate this section better into the revised paper **(lines 236–244)** and hope this will motivate readers not to skip this section. However, we would like to point out that the focus of the manuscript was decided sharply before writing the first version, and this focus is not promoting the shared stream-power model.

*L25: If authors search for more literature, there is a much wider range of concavity index found in nature.*

In previous papers, we referred to the range found in nature. Here we did not since it would be misleading with regard to the following sentence. The concavity of real rivers is affected, e.g., by inhomogeneous uplift. So a large scatter in values of $\theta$ cannot be transferred to the ratio $\frac{m}{n}$, while readers might think so if a range for $\theta$ was given. Beyond this – what would be the added value of this information?

*L35-39: This part needs to be rewritten in a much comprehensive manner.*

While we agree that the introduction did not describe the scaling issues sufficiently, we do not see why this part must be extended. This part has changed a bit **(lines 109–114)** since we restructured the introduction, but we have no idea what else we could write about this topic.

*L40: the linear diffusion equation would need a citation*

We added a reference **(line 114)**, although there is little added value.

*Eq(3): This is a key governing equation in this study, and it requrires much stronger justification. It also requires relevant literature.*

All literature about the shared stream-power model was already referenced in the first version. And since this paper is not about promoting this model, we prefer not to repeat the considerations of these papers completely. It should be sufficient that it is mathematically equivalent to two other models, which are at least partly accepted by the LEM community.

*Figure 1: I was very confused when I first looked at the figure. I guess what authors mean on the x-axis is the 'channel forming area', not 'catchment area'?*

Right, but we are quite sure that looking at the caption or at the label of the y-axis solved your confusion. Anyway, "channel forming area" is a good option here (also in the new form of the figure, where we had to switch to a cumulative plot for a better illustration of the scaling properties). We adopted this term at several occasions in the revised version **(lines 180, 187, 200, 209, 226–233, ...)**.

**Reviewer 4**

*This paper proposes an alternative solution to identify the channel – hillslope domain in dynamic landscape evolution models. The topic is of interest to the community. In general, the authors can give somewhat more depth to this story by pointing out issues they generally declare and by supporting their statements with literature and examples. Also, the results and findings would benefit from a clearer description at several points.*

*Title: I do not find the title to be adequate. This paper is not about self-organization of channels or hillslopes but rather presents a new LEM, that is essentially a full-scale fluvial model where hillslopes are represented as a drainage area independent process. There is no backup of any of the findings by field observations and the authors declare themselves that more research is needed to underpin this work and potential consequences. Hence, I would suggest a more technical title like: "A new approach to delineating channels in Landscape Evolution Models."*

Sorry, but the self-organization of channels and (not or!) hillslopes is the key result of this study. It is not about promoting a specific model for channel and hillslope erosion. **We added "and its potential for solving scaling issues" to the title** in order to point out more clearly what it is good for. However, we will not reduce the title to the scheme for delineating channels.

*Line 40. What do you mean with 'a scaling problem'? Please specify. Model components like SPACE (Shobe et al., 2017) have been used in combination with diffusion (Shobe et al., 2017).*

We hope that the existence of the scaling problem becomes clearer with the extended description in Sect. 1 **(lines 119–139)** and the example of diffusion in Sect. 2 **(lines 218–233)**. Just as a remark: Modular systems such as Landlab bring a great potential. And for a developer of a component such as SPACE it is tempting to write that it can be coupled to another component such as diffusion. But who takes care that it really works (not only technically)?

*In theory, all processes should act everywhere on a landscape. Why would diffusion as a process not act over channels and vice versa for fluvial incision? Naturally, at small discharges (drainage area) diffusion would be dominant over fluvial processes. I have been asking myself this question at several points throughout the manuscript and find it critical to address this point. Referring to other work does not suffice since this assumption is at the heart of this story.*

Right, but the question is whether the specific model is correct. To our experience, applying diffusion to all sites is not a problem. However, applying an erosion model that was made for channelized flow to hillslopes causes problems in the transition zone, although fluvial erosion vanishes for $A \to 0$. We hope that the extended description in Sect. 1 **(lines 119–139)** and the example of diffusion in Sect. 2 **(lines 218–233)** also clarifies this aspect.

*Line 63 Add SPACE (Shobe et al., 2017)*

SPACE is already promoted quite much by parts of the LEM community and would not fit so well into the line of studies cited here. Anyway, we found another place in the manuscript to cite it **(lines 54–56)**.

*Line 70 explain $K_d$ and $K_t$*

The description of the shared stream-power model was restructured a bit, so that the explanation of $K_d$ and $K_t$ is closer to their first occurence now **(lines 74–81)**.

*Line 125 here diffusion is applied to the entire domain. Just curious how the afore mentioned scaling issues are altering the results here. Aha, it is mentioned in the next sentence I see. Still wondering what those scaling issues are.*

We expanded this example by a comparison of different values of $D$ **(lines 218–233)** and hope that this will be useful for clarifying the scaling issue.

*Also, is the D value dimensionless? How does this compare to actual diffusion values ? (e.g. m2/yr see e.g. (Godard & Tucker, 2021))*

Ok, we added a statement that nondimensional properties are used for all simulations **(lines 177–178)**. Very recently, reviewers criticized the use of nondimensional properties in combination with a few additional metric examples that were intended to give a feeling for the orders of magnitude. So we decided not to use any dimensional values in this paper. Anyway, with $K = 2.5$ Myr$^{-1}$ and a grid spacing of 63 m, $D = 1$ would be the "typical" 10 m$^2$kyr$^{-1}$. For 6.3 m grid spacing, however, $D = 100$ would be 10 m$^2$kyr$^{-1}$.

*Line 150: energetically favorable means less energy, right? Maybe specify to help the readers a bit here.*

Right, we added a note **(line 253)**.

*Line 155. "In turn, we need a model for hillslopes that does not favor dendritic networks energetically" Not sure I understand why not, please explain better.*

We added one more sentence for explanation **(lines 255–258)**. However, we are not completely sure what the problem is and thus also not sure whether this sentence helps.

*Line 171: This might be true for the shared stream power model, but in the Carretier solution, a threshold slope is still used to calculate transport lengths. Please specify what you mean exactly.*

You are completely right – only the erosion part is the same, while the transport length is nonlinear in the approach of Carretier (2016) and also plays an important part for the occurrence of straight slope. So we removed this paragraph **(lines 275–281)** and return to nonlinear diffusion later **(lines 512–513)**.

*Line 209. The river is shorter, where? Explain better.*

We added a short explanation **(lines 321–323)**.

*Line 210: belongs*

Thanks! We fixed it **(lines 324–325)**.

*Line 211: 5000. How do we see that on the figure? Catchment A only goes up to 400 (dimensionless?)*

Yes, nondimensional properties are used throughout the paper. We guess that you were confused by the colorbar. This colorbar is optimized for recognizing the transition from hillslopes to channels ($0 \leq A \leq 400$), while the size of the entire catchment is much larger ($A = 5000$). This size can be estimated roughly from the range on the $x_1$ and $x_2$ axes. However, the size of the catchment is only an additional information. If you prefer, we remove it.

*Figure 3: Explain in the subscript what $A_h$ is. Makes the figure readable on itself.*

We added an explanation to the caption (now **Fig. 4**). However, we are not convinced that there is any way to make it readable without reading the main text.

*Figure 3–6: Are all these findings for nondimensional values/axes? Please specify.*

Ok, we pointed out more clearly that nondimensional properties are used for all simulations **(lines 177–178)**.

*Line 236: "Owing to the dominance of parallel flow patterns at hillslopes": That is interesting. So, at $A < A_h$, flow patterns do not organize in 'energetically favorable' patterns? Would be good to elaborate a bit on this.*

Ok, but better not at this point since we only refer to the consequence of parallel flow patterns here. We tried to elaborate the topic of dendritic vs. parallel flow patterns a bit more in Sect. 3 **(lines 255–264)**. Hopefully, it becomes clearer now that parallel flow patterns are in fact energetically favorable at the hillslopes. We really hope that it is clearer now since this is our key result on self-organization.

*244: Again, it has never been explained clearly what the scaling issues and such problems are. This is critical to support the value of this work. It does not suffice to point to previous work.*

This should hopefully be clearer now.

*Line 270: Would the authors expect differently when m/n is not 0.5?*

We performed some more tests, but (as expected) we did not find any exciting differences. The respective figures for $\theta = 0.25$ and $\theta = 0.75$ are shown in the responses to Reviewer 2. Overall, $\theta \neq 0.5$ is just more complicated, but there is no fundamental difference. Anyway, it was worth testing it in order to be sure. We added a few sentences **(lines 395–397)**.

*Line 272: I find these kinds of sentences of very little added value. I have no clue what is meant here unless I go read this paper. Either explain what is meant or drop the sentence.*

We added some explanation **(lines 401–404)**.

*Line 278: This paragraph needs some more context to be of added value for the paper. Is the focus on slope breaks, or rather on the orientation of streams? I was expecting to read how this model adjusts the SA plot one expects to see based on observations where a transition from a hillslope domain into a debris-flow dominated into a alluvial channel domain occurs (Montgomery & Foufoula-Georgiou, 1993). Please elaborate on that. Do we not see any hillslope domain because the model is actually a fluvial incision model where hillslope erosion does not depend on A? Curious to know.*

Of course, this section is about explaining why there is a break in slope from the hillslopes to the rivers. So it is rather designed for readers who want to understand why the approach works the way it does than for users. About the slope-area plot: We would, of course, see the hillslopes in the slope-area plot (Fig. 13, new numbering) if we plotted them. They would be a horizontal line with some "noise", depending on the model. We are afraid that you expect something completely different with regard to the debris-flow dominated regime. But should a distinct debris flow domain arise in the channelized domain without including debris flows explicitly in the model? To be honest, we have no idea about this point.

*Paragraph 8. As the authors seem to suggest this conceptual model seems to be disconnected from reality. Hence, it should be made clear what exactly the added value of this approach is. Why would one favor this method rather than just assuming continues processes of diffusion and incision (the latter maybe with an incision threshold)? If I would be to use a LEM; I am not convinced I would consider this approach in the way it is described now. Please summarize the benefits of this versus other approaches (other than the $A_c$ method).*

We are not completely sure what "disconnected from reality" means here. Is it just the the scheme for delineating channels does not work well for "noisy" real-world DEMs? Anyway, we added a new section (**Sect. 8**), which hopefully shows that the advantages are much bigger than the limitations.

*It would also be good to connect this work to field observations. Yes, it does not work well in its current state, but are there ways to improve this?*

These are two different aspects. Field obervations would preferably be slope breaks between hillslopes and channels in order to test the overall concept. This would be a nice work for a student. Extending the scheme for delineating channels towards "noisy" DEM would be another story. We have some ideas, but we are not sure whether we would arrive at an approach that is better and simpler than existing schemes.

*On a similar note: the authors show different simulations with various values of $A_h$. Are those values chosen arbitrarily? Can they be set using data or by using DEM-derived topographic metrics?*

We are quite sure that you will immediately find the answer if you imagine how large hillslopes would be for $A_h \ll 10$ and how many rivers would be there for $A_h \gg 1000$. And in order to clarify how to determine $A_h$ for applications, we added some more notes that $A_h$ is not an independent parameter, but derived from the parameters of the erosion models. So it would even not occur explicitly in model applications. Of course, we can use that result that channelization typically starts at $A = 2A_h$ for $m = 0.5$ and $n = 1$ in oder to constrain the other parameters bettter if we know at which catchment size channelization should start.

*Line 384. 'Serious problems'. That sounds a bit suspicious. Explain what the problems are and why they are assumed to be not seriously affecting model behavior.*

We tried to adjust it in such a way that it is also clear for those readers who only read the abstract and the conclusions **(lines 570–571)**.

*Line 389: What does it mean, works quite well?*

Do you seriously expect us to repeat all results here?